Distribution of bacterial communities along the spatial and environmental gradients from Bohai Sea to northern Yellow Sea

Yu Shu-Xian 1 2
Pang Yun-Long 1 2
Wang Yin-Chu 1
Li Jia-Lin 1
Qin Song sqin@yic.ac.cn 1
1 Key Laboratory of Coastal Biology and Biological Resource Utilization, Yantai Institute of Coastal Zone Research, Chinese Academy of Sciences , Yantai , Shandong , China
2 University of Chinese Academy of Sciences , Beijing , China
Hesham Abd El-Latif
Electronic publication date: 2018 Jan 29
Publication date: 2018
Volume: 6
Electronic Location ID: e4272
Received 2017 Aug 29; Accepted 2017 Dec 28
Copyright: ©2018 Yu et al.
Copyright year: 2018
Copyright holder: Yu et al.
License: This is an open access article distributed under the terms of the Creative Commons Attribution License, which permits unrestricted use, distribution, reproduction and adaptation in any medium and for any purpose provided that it is properly attributed. For attribution, the original author(s), title, publication source (PeerJ) and either DOI or URL of the article must be cited.
License URL: https://creativecommons.org/licenses/by/4.0/

Keywords: High-throughput sequencing, Bacterial biogeography, Geographic distance, Depth, Physicochemical conditions

Funding: Chinese Academy of Sciences strategic pilot project XDA1102040300 National Natural Science Foundation of China 41406192 41176144 41376139 National Key R&D Program of China 2016YFE0106700 Ocean Public Welfare Scientific Research Project State Oceanic Administration of China 201205027 This work was funded by Chinese Academy of Sciences strategic pilot project (XDA1102040300), National Natural Science Foundation of China (41406192, 41176144 and 41376139), National Key R&D Program of China (2016YFE0106700), and Ocean Public Welfare Scientific Research Project, State Oceanic Administration of China (201205027). The funders had no role in study design, data collection and analysis, decision to publish, or preparation of the manuscript.

==============================
The eutrophic Bohai Sea receives large amount of suspended material, nutrients and contaminant from terrestrial runoff, and exchanges waters with the northern Yellow Sea through a narrow strait. This coastal region provides an ideal model system to study microbial biogeography. We performed high-throughput sequencing to investigate the distribution of bacterial taxa along spatial and environmental gradients. The results showed bacterial communities presented remarkable horizontal and vertical distribution under coastal gradients of spatial and environmental factors. Fourteen abundant taxa clustered the samples into three distinctive groups, reflecting typical habitats in shallow coastal water (seafloor depth ≤ 20 m), sunlit surface layer (at water surface with seafloor depth >20 m) and bottom water (at 2–3 m above sediment with seafloor depth >20 m). The most significant taxa of each cluster were determined by the least discriminant analysis effect size, and strongly correlated with spatial and environmental variables. Environmental factors (especially turbidity and nitrite) exhibited significant influences on bacterial beta-diversity in surface water (at 0 m sampling depth), while community similarity in bottom water (at 2–3 m above sediment) was mainly determined by depth. In both surface and bottom water, we found bacterial community similarity and the number of OTUs shared between every two sites decreased with increasing geographic distance. Bacterial dispersal was also affected by phosphate, which was possible due to the high ratios of IN/IP in this coastal sea area.

Introduction

Microbes are recognized as the vital biological engines that drive global biogeochemical cycling (Falkowski, Fenchel & Delong, 2008). To better understand the microbial roles in biogeochemical processes, an indispensable subject is to unravel the mechanisms of microbial organization and succession over space and time in natural ecosystems (Van der Gast, 2013). According to Martiny et al. (2006), microbial biogeography can be determined by the influences derived from both contemporary environmental conditions and historical contingencies, which correspond to the active and passive reactions of microorganisms facing ever-shifting environment and geographic separation. The relative influence of environmental factors and dispersal limitation brought to bear onto microbial diversity is likely dependent on spatial scale (Martiny et al., 2011). At small spatial scales within a few kilometers, environmental conditions are frequently detected as the determinant forces affecting microbial composition, while the distance effects seem to be negligible; geographic separation can overwhelm any environmental effect at global scale; within an intermediate scale of ten to thousands of kilometers, both environmental factors and geographic distance are likely detected to affect community composition (Martiny et al., 2006). However, actual situations could be very complicated in natural ecosystems, because effects of local environment and historical factors are not always possible to be separated clearly (Lindstrom & Langenheder, 2012). Particularly in coastal waters, natural ecosystems are subject to intense interactions between terrestrial and oceanic processes, as well as frequent disturbance from human activities (such as pollution and eutrophication), resulting in regional heterogeneity of physicochemical conditions (Crossland et al., 2005). When physicochemical factors are spatially autocorrelated, spatial effects may superficially mask environmental effects. However, persistent environmental heterogeneity is in fact possible to overwhelm geographic separation in generating and maintaining microbial diversity at intermediate spatial scales (Wang et al., 2015).

In addition, water depth is another noteworthy factor that can shape vertical distribution of microbes in coastal seas (Zinger et al., 2011). Gong et al. (2015) have identified that water depth was the decisive factor to shape biogeography pattern of benthic microbial eukaryotes in the surface sediments of coastal sea. Our previous study (Li et al., 2014) also displayed that water depth played a surprisingly significant role in shaping bacterial biogeography in the upper 5 m of water column characterized with complex circulation systems. We consider that water depth may function as a comprehensive proxy of multiple physicochemical parameters rather than directly affect microbial diversity. Furthermore, coastal sea is usually defined as the seaward area with water depth <200 m (Pernetta & Milliman, 1995); such shallow depth may allow microbes frequently exchange through water column, and then contribute to microbial ecological shift in both surface and bottom layers of coastal waters. However, this issue could not be confirmed due to the lack of samples derived from bottom layer in our previous study.

Therefore the driving forces, including physicochemical factors, geographic distance and water depth, should be thoroughly considered when investigating microbial biogeography in coastal waters. Owing to pervasive applications of molecular approaches, especially high-throughput sequencing, plenty of studies have been conducted to reveal microbial biogeography and to compare the relative influences of above major forces in coastal surface water and sediments (Chamberlain et al., 2014; Gong et al., 2015; Langenheder & Ragnarsson, 2007; Lindstrom & Langenheder, 2012; Xiong et al., 2014). In contrast, less is known about the different relative contributions of major driving forces to microbial diversity in under-surface layers of shallow coastal water (seafloor depth <200 m).

Here we choose the sea area from Bohai Sea to northern Yellow Sea (Fig. 1) to investigate the spatial distribution of bacterioplankton. The Bohai Sea is an almost-enclosed sea area, where the average depth is 18 m and the water residence time is about 11–12 years (Lee et al., 2002). In contrast, the Yellow Sea is relatively a more open region located on the continental shelf, with an average depth of 44 m and a residence time of 5–6 years (Lee et al., 2002; Zheng, Fang & Song, 2006). The Bohai Sea and Yellow Sea can exchange sea water through a sole narrow strait. The study area has distinct seasonal characteristics; in summer, most of this area is clearly stratified, while the northwesterly monsoon destroys density stratification in winter (Lin et al., 2006; Su, 2001). Additionally, the Yellow River discharges a large amount of sediments and contaminants into the southern part of Bohai Sea. In 2008, about 19 × 103 tons of ammonium from Yellow River were loaded into the Bohai Sea, causing an average N/P ratio of 67 in the entire Bohai Sea (Dang et al., 2013). Increasing N/P ratio has been considered as an important factor to determine the dominant phytoplankton composition in the Bohai Sea shifting from diatoms to dinoflagellates during the last decades (Wei et al., 2004). In general, physicochemical conditions, e.g., temperature, salinity and nutrients, exhibit gradient distribution patterns in the surface and bottom water from Bohai Sea to northern Yellow Sea (Chen, 2009). Accordingly, this coastal region is an ideal model system to study the microbial distribution along spatial and physicochemical gradients, as well as to unveil relative influences of water depth, geographic distance and environmental factors. Therefore, our specific goals were to determine: (i) whether the dominant composition of bacterial communities varies along spatial and physicochemical gradients in this region; and (ii) whether geographic distance and depth contribute more than environmental factors to the spatial variation of microbial communities in both surface and bottom water.

Figure 1 The map of sampling sites in this study.

Materials and Methods

Sampling and environmental parameters

The sampling sites were determined along the center axis from the edge of Yellow River Estuary to northern Yellow Sea. Water collection was performed at seven stations (Fig. 1) during the open research cruise of the Bohai Sea and northern Yellow Sea in August/September of 2014. At each site, two liters of seawater was collected from the surface layer (0 m) and the bottom layer (2–3 m above sediment), and then filtered through 0.22 µm pore-size white polycarbonate filters. These filters were immediately sealed in sterile 50 mL centrifuge tubes and stored at −80 °C for further DNA extraction.

Environmental parameters, including temperature, salinity and depth, were measured in situ using a CTD sensor (Sea-Bird Electronics Inc., Bellevue, WA, USA). We measured pH, turbidity, dissolved oxygen (DO) and chlorophyll a (Chl a) on boat with a probe (Hydrolab MS5; HACH, Loveland, CO, USA). Concentrations of ammonium, nitrite, nitrate, phosphate and silicate were measured by a nutrient flow analyzer (Seal, Norderstedt, Germany).

DNA extraction, PCR amplification and high-throughput sequencing

The total DNA on filters was extracted using the FastDNA SPIN Kit for soil (MP BIO, Santa Ana, CA, USA) according to the manufacturer’s instruction. The quality of the DNA extracts was determined by agarose gel electrophoresis. We used a NanoDrop 2000c spectrophotometer (ThermoFisher, Waltham, MA, USA) to measure the DNA concentration.

The universe prokaryotic primers 515F (5′-GTG CCA GCM GCC GCG GTA A-3′) and 907R (5′-CCG TCA ATT CCT TTG AGT TT-3′) were used to target the V4–V5 region of 16S rRNA genes. Triplicate PCRs were carried out by ABI GeneAmp 9700, using purified DNA as a template. The PCR process started with a pre-denaturation period at 95 °C for 5 min, followed by 35 cycles of denaturation at 94 °C for 30 s, annealing at 55 °C for 30 s and elongation at 72 °C for 40 s, and ended with a final extension at 72 °C for 7 min.

PCR products were purified by agarose gel electrophoresis with an AxyPrep DNA Gel Extraction Kit (Axygen, Corning, NY, USA), and then quantified with a QuantiFluor-ST Fluorometers (Promega, Madison, WI, USA). High-throughput sequencing was performed at Majorbio Co., Ltd. (Shanghai, China) using the 250-bp PE Illumina MiSeq sequencing platform.

Data processing and statistical analysis

Raw sequence profiles were processed using QIIME (Caporaso et al., 2010b), and analyzed as previously described by Wang et al. (2015) with several modifications. In brief, retained high-quality reads need to meet the criterions of Phred quality score >20 (Q20), no ambiguous bases (no N’s), and consecutive high-quality bases >80% of total read length. Chimeras were filtered using USEARCH (Edgar et al., 2011). Operational taxonomic units (OTUs) were generated for the remaining sequences based on a 97% sequence similarity with the UCLUST method (Edgar, 2010). The representative sequences were picked for each OTU, aligned using PyNAST (Caporaso et al., 2010a), and assigned into taxonomic groups against the Greengenes database v.13.8 (DeSantis et al., 2006). Construction of phylogenic tree was conducted with the program FastTree (Price, Dehal & Arkin, 2010). The OTUs assigned into Archaea, Chloroplast and unassigned taxa were excluded from the further analysis of bacterial communities, as were those accounting for less than 0.01% of total sequences (Bokulich et al., 2013). Eventually, a total of 598,261 clean reads and 481 bacterial OTUs were obtained from 14 samples.

To estimate biodiversity of bacterial communities, we used the core_diversity_analyses.py script to calculate alpha-diversity indexes and beta-diversity distance with an evenly sequencing depth of 10,000 reads per sample. Multiple indices for alpha-diversity of bacterial communities were generated, including observed OTUs, Good’s coverage, phylogenetic diversity, Chao1 and Shannon-Wiener index. Weighted UniFrac distance was calculated to reveal beta-diversity. Principal coordinate analysis (PCoA) was performed to visualize community dissimilarities (beta-diversity) using R v.3.3.1 (R Development Core Team, 2008).

The dominant phyla, which had a relative abundance >1% in at least one sample, were utilized when making heat maps and conducting hierarchical clustering analysis (HCA). The discriminant taxa from each major cluster (suggested by HCA) were determined by the LDA (least discriminant analysis) effective size (LEfSe), using Kruskal–Wallis sum-rank test (α = 0.05 and one-against-all comparisons (Segata et al., 2011). Pearson’s and Spearman’s correlations between variables were calculated by R program. One-way analysis of variation (ANOVA) and least significance difference test (LSD, 0.5 confidence threshold) were conducted to determine the environment and community dissimilarity between surface and bottom layers. Pairwise geographic distances were generated from the longitude and latitude of each site with R program. Mantel and partial Mantel tests (with 999 permutations) (Legendre & Legendre, 2012) were conducted to unveil the relationships between environment (physicochemical factors), geographic distance, depth and beta-diversity (weighted UniFrac distance). For all correlation tests in this study, Benjamini–Hochberg multiple test correction was performed to control the false discovery rate of P values. To reveal spatial separation, we simulated general linear model (GLM) in scatter plots of geographic distance, water depth, environmental variables and community similarity (calculated as 1–weighted UniFrac distance or numbers of pairwise shared OTUs).

Results

Spatial variation of physicochemical parameters

We measured environmental parameters and calculated Pearson’s correlations between physicochemical parameters and spatial factors (longitude, latitude, water depth and collection depth; Dataset S1). In surface water of P1, the measured turbidity and salinity were 3.19 ntu and 30.05 g L−1 (Dataset S1A), indicating that P1 was near to the Yellow River Estuary in comparison with previous records (Shen et al., 2010; Zhang et al., 2014). Longitude showed a positive relationship with water depth (ρ = 0.91, P < 0.01; Dataset S1B), which is consistent with the topographic feature that water depth generally increases from the shallow coastal site P1 to the northern Yellow Sea site K5. With increasing longitude and water depth, the sampling site was further from the shallow coast and estuary. Thus, the negative relationship between longitude, water depth and turbidity (ρ =  − 0.71, P < 0.01; ρ =  − 0.6, P = 0.02; Dataset S1B; Fig. S1) was reasonable to understand. In addition, salinity also presented remarkable spatial autocorrelation (longitude, ρ = 0.83, P < 0.01; Dataset S1B; Fig. S1). Distributions of ammonium, nitrite, nitrate and phosphate were not detected to be prominently related to longitude and latitude (Dataset S1B–S1D), whereas the ratios of inorganic Nitrogen to Phosphorus (IN/IP) ranged from 10 to 152, with an average value of 55 (Dataset S1A). Collection depth was significantly correlated with temperature (ρ =  − 0.91, P < 0.01), salinity (ρ = 0.68, P = 0.01), pH (ρ =  − 0.76, P < 0.01), nitrate (ρ = 0.6, P = 0.02), phosphate (ρ = 0.81, P < 0.01) and silicate (ρ = 0.77, P < 0.01), suggesting distinct environmental conditions between surface and bottom water. One-way ANOVA and LSD test confirmed the dissimilarity of temperature, pH, nitrite, phosphate and silicate in surface and bottom water (P < 0.05).

Bacterial dominant taxa and alpha-diversity

Most of clean sequences (98%) were assigned into bacterial taxa. We focused our analyses on fourteen taxonomic groups that are greater than 1% relative abundance in at least one sample. These groups included Alpha-proteobacteria (30.5%), Gamma-proteobacteria (19.2%), Bacteroidetes (16.1%), Cyanobacteria (10.5%), Actinobacteria (5.8%), Planctomycetes (3%), Firmicutes (2.6%), Rhodothermaeota (2.4%), Delta-proteobacteria (2.4%), Beta-proteobacteria (1.8%), Verrucomicrobia (1.4%), SAR406 (1%), Tenericutes (0.4%), Chloroflexi (0.4%).

Hierarchical clustering analysis (HCA) showed that these abundant phyla/classes can roughly classify samples into surface and bottom clusters, except for P1s and B8s (Fig. 2). Figure 2 showed that P1s and B8s were clustered with P1b and B8b, indicating great similarity in these samples. Given that location of P1 and B8 was near the Yellow River estuary (Fig. 1), we consider bacteria in surface and bottom water might be mixing because of the terrestrial runoff. Therefore, it would be better to classify samples from P1 and B8 as a separate cluster (Cluster II). To determine the significant taxa in three groups, a total of 118 bacterial taxa were processed by LEfSe, and 68 taxa were significantly discriminant clusters (Dataset S2). Our results showed (see Fig. S2 and Dataset S2) that the most significant taxa in P1 and B8 (Cluster II) were associated with Bacteroidetes (including class Flavobacteriia, order Flavobacteriales, family Cryomorphaceae and genus Fluviicola), Beta-proteobacteria, Tenericutes (including family Acholeplasmataceae), and order Oceanospirillales (including family Halomonadaceae and genus Candidatus Portiera). Abundant bacteria in the other surface samples (Cluster I) were related with Cyanobacteria (including genus Synechococcus), Rhodothermaeota (including genus Balneola), and order Rickettsiales. The other bottom samples (Cluster III) were dominated by Planctomycetes (including class Planctomycetia, order Planctomycetales, family Planctomycetaceae and genus Planctomyce), Delta-proteobacteria (including order Desulfobacterales, family Nitrospinaceae, genus Nitrospina and SAR324), SAR406, and order Bacillales (including family Bacillaceae). These discriminant taxa from three groups can be well clustered by their correlation coefficients with spatial factors and environmental parameters (Fig. 3). Generally, most of the discriminant taxa from Cluster II exhibited correlations with longitude, water depth, salinity and turbidity (|ρ| > 0.5, P ≤ 0.05; Dataset S3). Abundant taxa of Cluster I were linked to pH and turbidity (|ρ| > 0.5, P ≤ 0.05), while the dominant bacteria of Cluster III were not only related to these factors, but also strongly affected by collection depth, temperature, dissolved oxygen, nitrate, phosphate and silicate (|ρ| > 0.5, P ≤ 0.05; Dataset S3). Besides, 50 abundant taxa accounting for over 50% of total OTU abundance were evenly dispersed among three clusters. Samples from Cluster I–III shared 70% of all detected OTUs, and more than 89 % of all OTUs were shared between surface and bottom water samples (Fig. 4). This implied that samples collected in this study shared a large portion of the local bacteria species pool, although some abundant bacteria could be particularly dominant or exclusive in each cluster.

Figure 2 Relative abundance of abundant phyla/classes can classify all samples into distinct clusters.

The abundant taxa are defined with relative abundance >1% in at least one sample. The phylum Proteobacteria is shown at class level. Color legend shows the relative abundance of bacterial taxa.

Figure 3 Spearman’s correlations between the most significant taxa of each cluster (Fig. 2) and spatial factors, depth, physicochemical variables are shown in this heatmap.

Color legend shows correlation coefficients. More details are shown in Dataset S3. DO, dissolved oxygen; Chl a, chlorophyll a.

Figure 4 Venn plots exhibit the OTU numbers shared among Cluster I–III (A), and between surface and bottom communities (B).

Alpha-diversity indexes including observed OTUs, phylogenetic diversity, Chao1 and Shannon-Wiener index, were calculated to estimate richness and biodiversity of bacterial community (Table S1). Within entire dataset of 14 samples, four estimators showed strong negative/positive correlations with the physicochemical factors temperature, pH, turbidity, dissolved oxygen, nitrite and silicate (|ρ| > 0.5, P < 0.05; Table 1). One-way ANOVA indicated Chao1 and Shannon-Wiener indexes of bottom communities were obviously higher than those in surface water (P < 0.05; Fig. S3 and Table S1), which implied bacterial communities in bottom water were more diverse than bacterioplankton in surface water. This was consistent with previous studies (Liu et al., 2015; Zinger et al., 2011). The environmental parameter phosphate was linked to the Shannon-Wiener indexes of bottom bacterial communities (|ρ| > 0.5, P < 0.05; Table 1). As for bacteria in surface water, a general decreasing richness and diversity occurred from P1 to K5, along with decreasing turbidity (Fig. S3, Dataset S1C and Table 1); the parameters longitude, water depth and Chl a were also closely related with alpha-diversity (|ρ| > 0.5, P < 0.05; Table 1).

Table 1 Spearman’s correlations between spatial factors, physicochemical parameters and alpha-diversity indexes.

Correlation tests were performed separately using all samples, samples from surface water and bottom water. Significant correlations are shown in this table, and highlighted with bold type (P ≤ 0.05 after Benjamini–Hochberg multiple test correction). Correlations with |ρ| ≥ 0.5 are labeled with asterisks. Chl a: chlorophyll a.

Samples	Variables	Observed OTUs	Phylogenetic Diversity	Chao1	Shannon–Wiener	
All samples	Temperature	−0.62*	−0.61*	−0.51*	−0.47	
pH	−0.61*	−0.65*	−0.49	−0.66*	
Turbidity	0.70*	0.72*	0.70*	0.86*	
Dissolved oxygen	−0.71*	−0.70*	−0.66*	−0.79*	
NO2-N	0.76*	0.71*	0.78*	0.74*	
NO3-N	0.42	0.44	0.24	0.67*	
SiO3-Si	0.63*	0.66*	0.57*	0.74*	
Samples from surface water	Longitude	−0.79*	−0.79*	−0.75*	−0.93*	
Water Depth	−0.89*	−0.89*	−0.86*	−0.82*	
Salinity	−0.63*	−0.63*	−0.63*	−0.88*	
pH	−0.68*	−0.68*	−0.64*	−0.75*	
Turbidity	0.79*	0.79*	0.86*	0.93*	
Chl a	0.79*	0.79*	0.89*	0.5*	
Samples from bottom water	PO4-P	−0.40	−0.46	−0.29	−0.86*	

Microbial distribution in surface and bottom water

Using weighted UniFrac distance, we conducted PCoA for all samples. The PCoA plots suggested separation of bacterial communities from surface and bottom water (Fig. 5). The first three principal coordinates PC1, PC2 and PC3 explained 80.89% of the total variation; PC1 and PC2 (Fig. 5A), together accounting for 69.27% of the total variation, demonstrated a similar clustering of samples as that in Fig. 2.

Figure 5 Spatial variations of bacterial communities are exhibited by principal coordinates analysis (PCoA) with the coordinates PC1 and PC2 (A), and the coordinates PC1 and PC3 (B).

Samples from surface and bottom water are labeled with blue squares and red dots.

Table 2 Mantel and partial Mantel tests for the Spearman’s rank correlations between geographic distance, water depth, collection depth, environment and beta-diversity.

Tests were performed separately using all samples, samples from surface water and bottom water. Correlations are shown in this table and are labeled with an asterisk when |ρ| ≥ 0.5. When P values are less than 0.05 after Benjamini–Hochberg multiple test correction, correlations are shown with bold type. Environment was calculated as Euclidean distances of physicochemical parameters.

Samples	Variables	Mantel tests	Partial Mantel tests control factors	
			Geographic distance	Water depth	Collection depth	Environment	
All samples	Geographic distance	0.18	–	0.10	0.19	0.18	
Water depth	0.16	0.04	–	0.12	0.13	
Collection depth	0.44	0.45	0.43	–	0.35	
Environment	0.30	0.29	0.28	0.08	–	
Samples from surface water	Geographic distance	0.49	–	0.47	–	0.35	
Water depth	0.21	−0.13	–	–	0.26	
Collection depth	−0.08	0.07	−0.05	–	−0.01	
Environment	0.57*	0.47	0.58*	–	–	
Samples from bottom water	Geographic distance	0.52*	–	0.09	0.08	0.53*	
Water depth	0.78*	0.68*	–	0.17	0.76*	
Collection depth	0.77*	0.67*	−0.02	–	0.76*	
Environment	0.34	0.37	0.19	0.34	–	

Mantel and partial Mantel tests were performed to further explore the effects of geographic distance, spatial factors (water depth and collection depth) and physicochemical variables (environment) (Table 2 and Table S2). Our results showed that the beta-diversity of all 14 bacterial communities was correlated with collection depth and environment (P < 0.01; Table 2), while the distance effect was relatively weak (ρ = 0.18, P > 0.05). For the beta-diversity of bacterial communities in bottom water, strong effects were mainly derived from geographic distance, water depth and collection depth (ρ > 0.5, P < 0.05; Mantel tests), but the correlations between beta-diversity and geographic distance became insignificant when water depth or collection depth was controlled (P > 0.05; partial Mantel tests). Since collection depth of bottom water was determined as 2–3 m above seafloor, water depth could be regarded as a factor tantamount to collection depth in shaping community dissimilarity. Hence, water depth was the most important factor generating beta-diversity of bottom bacterial communities. Mantel tests indicated that the beta-diversity of bacterioplankton in surface water was correlated with geographic distance (ρ < 0.5, P < 0.05) and environment (ρ > 0.5, P > 0.05), while partial Mantel tests suggested their correlations were likely mutually dependent (|ρ| < 0.5, P > 0.05).

Effect evaluation of each physicochemical parameter revealed that temperature, pH, turbidity, dissolved oxygen, nitrite and silicate was the most important environmental factors influencing the beta-diversity of entire 14 bacterial communities (P < 0.05; Mantel tests in Table S2), further confirming the dissimilarity between surface bacterioplankton and bottom bacterial communities. Correlations between temperature, salinity, phosphate and beta-diversity of bottom bacteria (|ρ| > 0.5, P < 0.05; Mantel tests) were observed to be dependent on water depth and collection depth (P > 0.05; partial Mantel tests). As to the bacterioplankton in surface water, community dissimilarity was related to turbidity and nitrite (|ρ| > 0.5, P < 0.05; Mantel tests in Table S2) without any noticeable dependence (partial Mantel tests).

Figure 6 Relationships between geographic distance, water depth and numbers of pairwise shared OTUs are shown in this figure.

Scatter plots were generated separately for samples from surface (A–B) and bottom water (C–D). Regression lines, along with regression coefficients (R) and probability (P), were generated using general linear model (GLM).

Numbers of pairwise shared OTUs decreased with increasing geographic distance of every two sites in both surface and bottom water, as well as pairwise dissimilarity (Euclidean distance) of water depth, longitude, salinity and phosphate (Fig. S5, Fig. 6 and Table 3), which clearly suggested the dispersal of bacteria in connective waters were affected and limited by these factors.

Table 3 Spearman’s correlations between the numbers of pairwise shared OTUs and pairwise Euclidean distances of multiple variables.

Correlation tests were performed separately using samples from surface water and bottom water. Significant correlations are shown in this table and are labeled with an asterisk when |ρ| ≥ 0.5. When P values are less than 0.05 after Benjamini–Hochberg multiple test correction, correlations are shown with bold type.

Variables	Samples from surface water	Samples from bottom water	
Geographic distance	−0.72*	−0.67*	
Water depth	−0.87*	−0.87*	
Longitude	−0.74*	−0.69*	
Temperature	−0.21	−0.89*	
Salinity	−0.82*	−0.81*	
PO4-P	−0.55*	−0.49	

Discussion

Composition of the microbial communities distributed along spatial and environmental gradients

We observed horizontal salinity and turbidity gradients ranging from the Bohai Sea to northern Yellow Sea (Fig. S1), while gradients in temperature, salinity, pH, nitrate, phosphate and silicate were reflected in sample collection depths (Dataset S1). Under such condition, the dominant taxa of bacterial communities classified samples into three distinct clusters (Figs. 2 and 3), demonstrating patterns of bacterioplankton distribution among sampling sites.

For instance, genera Synechococcus and Balneola characterized Cluster I, predominated in surface bacterioplankton, and decreased with increasing collection depth (Fig. 3 and Dataset S3). Species in genus Synechococcus require sun light for primary production (Paerl et al., 2012); genus Balneola includes aerobic microbes or facultative anaerobes (Munoz, Rossello-Mora & Amann, 2016), such as Balneola vulgaris, which has been isolated from surface waters in coastal sea (Urios et al., 2006). Bacterial taxa discriminating Cluster I were likely adapted to surface water conditions as exemplified by the differential abundance of cyanobacterial OTUs and known aerobic heterotroph OTUs in surface samples.

Cluster II was dominated by Bacteroidetes (including class Flavobacteriia and genus Fluviicola), Beta-proteobacteria, Tenericutes (including family Acholeplasmataceae), and order Oceanospirillales (including genus Candidatus Portiera) (Fig. 3). Most of these bacterial taxa showed significantly negative correlationships with longitude (Dataset S3), suggesting the outspread effects from estuarial waters (Campbell & Kirchman, 2013; Liu et al., 2015). Among all these taxa, Flavobacteriia-lineage is related to algal organic matter degradation in the coastal water, which may facilitate the growth of Oceanospirillales (Williams et al., 2013); genus Fluviicola is usually retrieved from river water and riverbeds (Bowman, 2014). Our results showed family Acholeplasmataceae was mainly associated with genus Acholeplasma (Dataset S2), species of which are the most common mollicutes in vertebrate animals (Martini et al., 2014). Genus Candidatus Portiera is recognized as the primary endosymbiont of whiteflies (Thao & Baumann, 2004). In our study, these land-source bacteria were dominant and confined in sites P1 and B8, consistent with their close proximity to mouth of the Yellow River (Van der Gast, 2015).

Discriminant taxa in Cluster III were Planctomycetes (including genus Planctomyce), Delta-proteobacteria (including genus Nitrospina and SAR324), SAR406, and order Bacillales (including family Bacillaceae) (Fig. 3 and Fig. S2). These taxa were closely related to nutrients distribution (Dataset S3), because they are important participants in biogeochemical cycling of carbon, nitrogen, phosphorus and sulfur. For instance, phylotypes of genus Nitrospina are known as nitrite oxidizers, and have been detected in sediments of South Pacific (Durbin & Teske, 2011). SAR324 and SAR406 are typically deep-sea group, and found in aphotic ocean with low-oxygen conditions (Alonso-Saez, Diaz-Perez & Moran, 2015; Salazar et al., 2016); some of their species are involved in carbon cycling and sulfur oxidation (Sheik, Jain & Dick, 2014). Members of family Bacillaceae perform fundamental roles in degradation of organic matter, nitrification, denitrification, nitrogen fixation and phosphorus solubilization (Mandic-Mulec, Stefanic & Van Elsas, 2015). As our results indicated, Bacillaceae-lineage was closely related to pH, dissolved oxygen, nitrite, nitrate, phosphate and silicate (Fig. 3 and Dataset S3).

In summary, the dominant microbial taxa of each province were greatly affected by spatial control and environmental conditions, which was consistent with their physiological requirements, dispersal limitation and ecological functions. Meanwhile a considerable number of abundant taxa dispersed ubiquitously between surface and bottom layers, and among entire sampling sites (Dataset S2 and Fig. 4). As Stocker (2012) has reviewed, some microbes actively exploit environmental heterogeneity, whereas some others just adapt the dynamic physicochemical gradients. Moreover, each single site has been suggested having a persistent microbial “seed bank”, which can go through ever-shifting environmental conditions over time (Caporaso et al., 2012). Therefore in our study sea area, strong vertical mixing in winter could enhance passive dispersal of bacterioplankton through water column, as well as enlarge the opportunity for microbial seed banks to exchange their species. Those bacteria with broad niche breadths can widely spread and sustain in the sea water with the help of water movements. When it comes to summer, stratified seawater shaped distinct environmental conditions in this area, leading to thriving populations of some bacteria and distinct clustering of microbial communities, while the samples in surface and bottom (as well as Cluster I–III) still shared a large amount of OTUs (Fig. 4). We considered this could be an optimal strategy for bacterioplankton to maintain biodiversity and functions through annual cycles of mixing and stratification in sea water.

Different factors shaped the diversity of bacterial communities in surface and bottom water

Our results demonstrated geographic distance, spatial factors and physicochemical variables contributed significantly to the diversity of bacterial communities, and their relative influences varied in the surface and bottom communities. For bacterioplankton in surface water, both alpha- and beta-diversity had significant correlations with turbidity (Table 1 and Table S2). Distribution of turbidity was spatially autocorrelated (Dataset S1C); the higher turbidity occurred at P1 (Dataset S1B), partly because the Yellow River discharges mixed the surface and bottom water with large amount of sediments along with particles, nutrients and organic matters. High turbidity can cause decrease in light penetration, which explained the significant abundance of photosynthetic bacteria (Cyanobacteria) in surface water of B6-K5 rather than at the estuarial sites P1 and B8 (Fig. 2). In our study, Synechococcus-lineage was the dominant bacteria in surface water, and the most abundant Synechococcus OTU, denovo 4779, was 100% identical to Synechococcus sp. strain WH 8103 according to BLAST results. Relative abundance of Synechococcus-lineage negatively correlated with ammonium, nitrate and nitrite concentrations, reflecting these microbes preferring lower nutrients in the northern Yellow Sea (Dataset S3). Nitrogen availability (nitrate and nitrite) has been suggested as one factor determining the relative abundances of Synechococcus-lineage in temperate seawater (Choi, Noh & Shim, 2013), whereas our results showed that nitrite (ρ =  − 0.56) had a strongereffect than nitrate (ρ =  − 0.49, Dataset S3). Additionally, nitrite was another environmental factor greatly affecting beta-diversity of bacterioplankton (Table S2). According to the previous study (Bird & Wyman, 2003), assimilation of nitrate and nitrite in Synechococcus strains can be repressed by high concentrations of ammonium. For instance, ammonium at concentrations above 1 µmol L−1 can represses the expression of genes involved in nitrite/nitrate assimilation and inhibits the uptake of nitrate (Lindell, Padan & Post, 1999; Lindell & Post, 2001). However, Synechococcus sp. strain WH 8103 is capable of assimilating nitrite with the presence of high ammonium concentrations (Wyman & Bird, 2007). Here we measured the concentrations of ammonium ranged from 0.45 µmol L−1 to 1.61 µmol L−1 in the study area (Dataset S1A). This means nitrate uptake might be inhibited in some sites with ammonium concentrations >1 µmol L−1, while in all surface sites, nitrite assimilation was subject to less inhibition because of the abundant Synechococcus sp. strain WH 8103. This could explain the detected result that nitrite was more relative with the bacteria diversity than nitrate.

For bacteria in bottom water, community similarity was mainly determined by water depth and collection depth, presenting a depth-decay pattern (Table 2 and Fig. S4). Similar pattern has been observed in coastal and deep-sea sediments, where composition and structure of benthic bacterial community differed significantly with increasing water depth (Jacob et al., 2013; Zinger et al., 2011). In our study, water depth varies along the spatial gradient from the shallow coast to northern Yellow Sea. The shallower water depth was near the estuary with mixing of communities from bottom to surface (for example, P1 and B8). With increasing water depth, bacteria in bottom layer were farther from the effects of mixing in estuary. In vertical dimension, we consider collection depth works as a comprehensive proxy for many physicochemical factors (including temperature, salinity and phosphate, Dataset S1D) as reported previously (Fortunato et al., 2012; Gong et al., 2015; Li et al., 2014).

Sea water in the study area is subject to annual circles of winter mixing and summer stratification, and also affected three important water bodies, the Yellow River, Bohai Sea and Yellow Sea, which might lead to the large core bacterial species bank as well as the community distinctions existing in surface and bottom samples (Figs. 4 and 5). Our results showed both the surface and bottom bacteria were constrained and affected by geographic distance, longitude, depth, salinity and phosphate (Fig. 6 and Table 3). Phosphate did not exhibit dependence on the spatial factor longitude (Dataset S1); meanwhile the IN/IP ratios in most of samples were much higher than the Redfield’s ratio 16 (Redfield, 1958), suggesting a phosphate insufficiency for biological communities relative to inorganic nitrogen. Thus we consider that phosphate affecting bacterial dispersal might be unrelated to the spatial reason, and the most possible explanation should be the high ratios of IN to IP.

Conclusion

In this study, we observed significant differences of environmental conditions and a distinct separation of bacterial communities in surface and bottom waters. Vertical separation of bacterial communities has been reported in ocean by numerous studies (Fortunato et al., 2012; Qian et al., 2011; Treusch, 2009). This could be explained by stratification of water column as well as vertial distribution of multiple physicochemical factors, such as temperature, salinity, light and nutrients (Fortunato et al., 2012).

Now we may answer the questions driving this study: (i) do the abundant bacteria vary along spatial and physicochemical gradients? Yes. In our study, those discriminant taxa exhibited limited distributions in the study area, while the others presented a ubiquitous distribution. Bacterial distribution along the gradient from the shallow coast to relative open sea shows remarkable horizontal and vertical patterns in bacterial communities. (ii) Do geographic distance and depth contribute more than environmental factors do to spatial variation of microbial communities? The situation differs in bacterial communities form surface and bottom waters. Environmental factors significantly affected the composition and biodiversity of bacterial communities in surface water. In coastal sea, water depth plays a noticeable role in biogeography of bacteria in bottom water, and leads to a depth-decay pattern of community similarity. Geographic distance enhanced community dissimilarity as previously reported (Martiny et al., 2011), and was one of the most important factors driving the spatial variation of microbial communities. However, the only summer samples are not sufficient to uncover the specific mechanisms of their ecological function. To unveil more details, further study is still necessary to integrate time-series data and more complete environmental profile.

Supplemental Information

Supplemental Information 1 Supplementary materials

Click here for additional data file.

Dataset S1 Environment factors and their correlations

(A) Environmental parameters determined for each samples.

(B) Pearson’s correlations between environmental variables. Significant correlations are highlighted with bold type for probabilities (P) in lower triangle, and with gray shadow for coefficients (ρ ) in upper triangle.

(C) Pearson’s correlations between environmental variables in surface water. Significant correlations are highlighted with bold type for probabilities (P) in lower triangle, and with gray shadow for coefficients (ρ ) in upper triangle.

(D) Pearson’s correlations between environmental variables of bottom water. Significant correlations are highlighted with bold type for probabilities (P) in lower triangle, and with gray shadow for coefficients (ρ ) in upper triangle.

Click here for additional data file.

Dataset S2 Results from least discriminant analysis (LDA) effect size

Click here for additional data file.

Dataset S3 Spearman’s correlations between the significant taxa from Fig. 3 and environmental factors are shown in this table

Significant correlations (P ≤ 0.05 after Benjamini-Hochberg multiple test correction) are highlighted with gray shadow.

Click here for additional data file.

The authors acknowledge the editing of English by Prof. Kevin McCartney (University of Maine at Presque Isle) and Prof. Andrzej Witkowski (University of Szczecin).

Additional Information and Declarations

Competing Interests

Author Contributions

Data Availability

The authors declare there are no competing interests.

Shu-Xian Yu conceived and designed the experiments, performed the experiments, analyzed the data, wrote the paper, prepared figures and/or tables.

Yun-Long Pang and Yin-Chu Wang contributed reagents/materials/analysis tools, reviewed drafts of the paper.

Jia-Lin Li and Song Qin analyzed the data, reviewed drafts of the paper.

The following information was supplied regarding data availability:

The raw sequence data of this study have been deposited to the Sequence Read Archive of NCBI (https://www.ncbi.nlm.nih.gov/) under accession number SRR5312630–SRR5312643.

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
