# Peer review of "Distribution of bacterial communities along the spatial and environmental gradients from Bohai Sea to northern Yellow Sea"

_PeerJ, doi:10.7717/peerj.4272_

## Round 0.1 · original submission · Major Revisions

The manuscript requires a number of Major Revisions - please examine the reviewer reports and address their comments accordingly.

·

Basic reporting

This manuscript describes a field study designed to examine microbial biogeography in surface water and benthic boundary layer water in the Bohai Sea of coastal China. I am pleased that this contribution focuses on the microbial diversity in marine and brackish environments of China and broader east Asia as this region has traditionally received less attention in the marine microbial ecology/oceanography literature. I commend the authors for their hard work collecting this dataset and for also sampling multiple biogeochemical parameters in addition to collecting samples for sequencing. I also commend the authors on the well curated datasets/tables/figures submitted with this manuscript.

The writing of the manuscript is generally understandable, although there are many places (highlighted below) with awkward English phrasing and syntax coupled with rather simple grammar mistakes. The authors might benefit from having a native English speaker help them clarify certain sentences, remove unnecessary words, and ease overall comprehension. I have noted several examples below where the English writing could be improved to aid in reader comprehension.


Figure 2: “explained” is misspelled

Physiochemical is misspelled throughout text

Abstract:
Line 21: change “matters” to “material”

Line 24: replace “spatial” with “the”

Line 25: replace “communities” with “taxa”

Line 25: remove “the” before “spatial”

Line 26: “coastal gradients” Gradients of what? Please specify

Line 28: “sunlit surface layer” Unclear if this refers to the “shallow coastal water (depth < 20m)” or is a unique depth strata in addition to the bottom layer. Please clarify.

Line 33-34: “Geographic distance enhanced bacterial community dissimilarity and limited bacteria distribution in both surface and bottom water.” Unclear to me what is meant by this sentence... Are you saying that communities became more taxonomically dissimilar as geographic distance between samples increased? And that this trend held in both surface and bottom water? Please clarify

Introduction:
Lines 60-63: “Hence, spatial effects may superficially mask environmental effects when physicochemical factors are spatially autocorrelated, while in fact persistent environmental heterogeneity is possible to overwhelm geographic separation in generating and maintaining microbial diversity at intermediate spatial scales.” This sentence is unclear to me and should probably be split into two sentences. Are you saying that community composition is a function of both dispersion (so called spatial effects) as well as selection due to variable habitat conditions? And that this balance is effected by the scale at which you are defining your study area? As I understand you essentially have a two-factor study design where you must account for the effects of spatial separation/dispersion between similar habitats and combined niche adaptation+spatial separation differences between dissimilar habitats (i.e bethic boundary layer versus surface water). Again, I think this sentence should be clarified.

Line 63-66: “As Wang et al. (2015)…” sentence probably not needed

Line 71: “In the surface water” Please specify. For example, “in the upper 20 meters of the water column”

Line 85: In contrast instead of “by contrast”

Line 86: “shallow coastal water” Again please be specific about depths

Line 91: located instead of “locating”

Results:
Line 197: Strange wording... I suggest instead "Bacterial diversity and abundance"

Lines 198-199: “Thereinto, fourteen phyla including the main classes of phylum Proteobacteria were abundant …” This sentence has unusual wording… Try instead “We focused our analyses on fourteen taxonomic groups that were greater than 1% relative abundance in at least one sample. These groups included the Phyla … and the classes Alphaproteobacteria, Gammaproteobacteria, Betaproteobacteria, and Deltaproteobacteria.”

Line 205: confusing syntax “This indicated the abundant taxa of P1 and B8 distributed in surface and bottom water with great similarity.”

Line 209: replace “taxa of them were found to be the discriminant taxa in these distinct clusters” significantly discriminant clusters

Line 223: remove “remarkable negative/positive”

Line 226: “III were also strongly” Do you mean Cluster II here?

Line 227: remove “Despite of these discriminant taxa”

Line 227: “Despite of these discriminant taxa, there were still 50 abundant taxa, whose relative abundance together reached over 50 % in each cluster, dispersed indistinguishably in three clusters” Unclear wording... I'm not actually sure what you mean. Do you mean "50 taxa accounting for over 50% of total OTU abundance were evenly distributed amongst all three clusters."?

Line 229: “Moreover, Cluster I-III shared 70 % of total OTUs, and the shared OTUs between surface and bottom bacterial communities accounted for over 89 %” Again unclear wording/syntax/grammar… I'm not entirely sure what you are saying here, but I think it would be clearer if you wrote "Samples from clusters I-III shared 70% of all detected OTUs. Furthermore, 89% of all OTUs were shared between surface and bottom water samples."

Line 247: remove “Driving forces of”

Line 248: Please be explicit about the input data for your PCoA analysis.

Line 248: Replace “exhibited clear clusters” with suggest separation

Line 270: remove “striking”

Line 271: remove “close”

Line 271: “correlationships” should be correlations

Line 276: remove “intriguingly”

Line 276: “geographic distance” Unclear - geographic distance of what? Do you mean pairwise physical distances between two samples? Please specify

Line 248: Replace “exhibited clear clusters” with suggest separation

Line 270: remove “striking”

Line 271: remove “close”

Line 271: “correlationships” should be correlations

Line 276: remove “intriguingly”

Discussion:
Line 283: remove “Generally, environmental variables presented remarkable spatial gradients.”

Line 283: “Salinity and turbidity were related to longitude, shaping horizontal gradients from the Bohai Sea to northern Yellow Sea; collection depth led to vertical dissimilarity of temperature, salinity, pH, DO, Chl a, nitrate, phosphate and silicate between surface and bottom water (Dataset S1).” Again, rather unclear wording and phrasing... I suggest something like "We observed horizontal salinity and turbidity gradients ranging from the mouth of the Yellow River to the northern Yellow Sea while gradients in temperature, salinity, pH, dissolved oxygen, Chl A, nitrate, phosphate and silicate were reflected in sample collection depths."

Line 288: Replace “non-random distribution pattern of bacterioplankton” with patterns of bacterioplankton distributions.

Line 295: replace “actually” with likely

Line 295: “to their physiological requests for adequate light and oxygen, as well as the other beneficial conditions in surface water.” I don't think you have enough evidence to support this statement as it is written. I would say something like "Bacterial taxa discriminating Cluster I were likely adapted to surface water conditions as exemplified by the differential abundance of cyanobacterial OTUs and known aerobic heterotroph OTUs in surface samples.

Line 309: Replace “the microbial hypothesis that some microorganisms are biogeographically restricted.” with their close proximity to mouth of the Yellow River.

Line 369: “This could explain why nitrite was more relative with the bacteria diversity than nitrate in our results.” This statement needs to be revised for proper English grammar. I cannot decipher its meaning as it currently stands.

Experimental design

I think the research question, motivations, and description of the methods is sufficient for publication in PeerJ.

Validity of the findings

The results appear to be derived from sound sampling, processing, and sequencing methodologies, and for the most part (with some exceptions noted below) the statistical analyses appear to be robust. The conclusions are not overstated, appear to be adequately supported, and are largely consistent with the oceanographic features of the region (again I have some additional recommendations below).

Minor Comments:
Figure 1: It would be useful to have a figure inset displaying where the Bohai sea is located with respect to the rest of coastal China. This would give extra context for readers who are not familiar with the region.

Instead of using terms like “surface waters” I would recommend being more specific and choosing something like “the upper 5 meters of the water column,” “the upper 20 meters” and so on.

When mentioning N/P ratios please be more specific if you are talking about total nitrogen or inorganic nitrogen.

DO should be replaced with “dissolved oxygen” throughout the text.

Results:
When discussing correlations between variables in this section please write the correlation and the P value.

Line 248: Please be explicit about the input data for your PCoA analysis.

Line 276: “geographic distance” Unclear - geographic distance of what? Do you mean pairwise physical distances between two samples? Please specify

Discussion:
Line 381: “Additionally, adaption to different pressure might also explain the bacterial community variation in bottom water.” I suspect that pressure differences are not driving the patterns in your study. Pressure is a relevant parameter at great depths (over 1 km) but I don’t really think it would be important at depths of < 100 meters such as in your study.

Line 387: remove “Unlike geographic distance, depth and salinity,”

Major Comments:
Please explain your rationale for assigning sample clusters in this this study. More specifically, how did you classify clusters in Figure 2? Why is B8b classified as Cluster II and not Cluster III? I see strong evidence in Figure 1 for two well-supported clusters but is not obvious to me that Clusters II and III should be separated.

Figure 3: How do you define “most significant taxa of each cluster” are these from the results of your LEfSe analysis? Why did you not include all the discriminant taxa in Fig S1? This should be briefly explained in the figure legend.

Please be precise and clear when discussing sample collection depth and overall water depth, otherwise this terminology can be confusing.

I would recommend trying to distill your tables down to the most relevant correlations for your story. It will help your reader grasp the main relationships presented in your study. For example, “latitude” has no significant correlations in Table 1 at all. I also recommend omitting the P value column altogether and simply only highlight a correlation if it exceeds 0.5 and has a P value < 0.05. Then mention this in the table legend. For me P values are rarely interesting in and of themselves, but that is my own preference.

The structure of Table 1 and Table 2 is somewhat confusing to me. What I think you are intending to convey is that the correlation for “Longitude, ect” is due to the alpha diversity index calculated for all OTUs at that sampling station while the correlations underneath “Longitude, ect” denoted by “surface” and “bottom” refers to correlations for alpha diversity indices calculated only for the surface and bottom depths respectively. Is this correct? If so, you should explain this more clearly in the table legends. Additionally, it is unclear why you need a separate correlation analysis for Wat-Depth and Col-Depth… Indeed, the “bottom” correlation for the two categories are identical and there is no comparison for surface, which I am assuming is because all your surface samples have the same collection depth… But (assuming my above understanding is correct) then why does it make sense to have an integrated surface+bottom Col-Depth comparison here? Again, I think this needs some clarification… Please see my comment above.

Table 2: Please clarify what your distance matrix inputs are for your mantel tests. I am assuming you are performing the mantel test on UniFrac distances versus a distance matrix of pairwise sample distances in each category (Geo-Distance, Wat-Depth, ect) while controlling for the effects of each category in your partial mantel tests. This information would be useful in a table legend.

Lines 388-392 (Conclusion): “meanwhile the N/P ratios in most of samples were much higher than the Redfield's ratio 16 (Redfield 1958), suggesting a severe phosphate insufficiency for biological communities relative to inorganic nitrogen.” You discussed the N/P ratio little in the results, yet conclude with a statement referencing it which does not seem logical to me. N/P should either be mentioned more in the discussion prior to this sentence. Here I suspect the deviation of the measured N/P ratio from Redfield ratios is due to terrestrial inputs – probably from local agriculture. I find it unlikely that these communities would be experiencing “severe” P limitation, but rather they just have a large excess of N relative to P due to riverine inputs resulting from human land use. This distorts the N/P ratio from Redfield, but doesn’t really imply anything about limitation. I think you would need to do a number of additional experiments to claim “P limitation.”

Lines 164-165: (Methods) “Pearson’s and Spearman’s correlations between variables were calculated by R program.” When conducting many statistical tests (in this case correlation) one should correct P values for multiple testing artefacts (ie false positives simply due to the increasing number of comparisons). This can be done using the conservative Bonferroni Correction, but I would recommend using the Benjamini-Hochberg multiple test correction which controls the false discovery rate. Did you use multiple test correction? If so you should include this in your methods.

Reviewer 2 ·

Basic reporting

Overall, the article is clear and well written. However, there are enough small grammatical errors that I think the authors would benefit from having it edited by a native English speaker in order to avoid any ambiguity. The figures are well constructed and add to the overall understanding of the text. I do think, however, that a figure perhaps in the form of a few section plots of important environmental parameters across the sampling region would greatly benefit the understanding of the text and give the reader a better sense for the study area. The supplemental data and figures are accessible and clear to understand. I find Figure S4 highly valuable, however, it is slightly confusing because the color scheme used to identify each of the three regions are the same colors used repeatedly in the Venn diagrams to represent different stations. Consider using a different color scheme to identify each region.

Experimental design

The study goals are clearly stated in the introduction as well as the motivation for undertaking the study specifically in this region. The methods are well described and within the current standards for the field. I think seeking to understand the factors that determine the composition of bacterioplankton in the marine environment as well as factors that may limit their dispersal are important factors to consider. One of the main questions posed by the authors is whether geographic distance/depth contribute more than environmental factors to the structure of bacterial communities. However, truly separating the effects of these factors is difficult as many of them co-vary together. And the authors acknowledge in their results that their statistical analysis indicates they are at least partially dependent on each other, making their question difficult to clearly answer. I think the authors could benefit by basing their analysis on the physical structure of the water column across their study region and using a water mass analysis rather than simply distance/depth to determine what may be more important in structuring bacterial community composition and potentially limiting dispersal. Because the study region contains three distinct bodies of water (the Yellow River, the Bohai Sea, and the Yellow Sea) I think this would be feasible and allow the authors to make clearer conclusions.

Validity of the findings

The authors perform a rigorous statistical analysis of their data to help support their findings and very clearly address their first posed question. As stated above, I think their analysis for the second question posed could be strengthened by basing their analysis on a physical structuring of the water column and investigating the influence of various water masses and mixing processes rather than geographic distance and depth which co-vary with many environmental factors.

---

## Round 0.2 · accepted · Accept

Dear Sir,

Your manuscript has been Accepted for publication

·

Basic reporting

The writing has been properly clarified in response to most of my comments. The authors have done well at improving the English writing and generally increasing the clarity and depth of explanation for many sections.

Experimental design

Thank you for including multiple test corrections in your statistical analysis.

Validity of the findings

no comment.